Brief Communication

# TemplateFlow: FAIR-sharing of multi-scale, multi-species brain models

Rastko Ciric [1,2] ✉, William H. Thompson[1,3,4], Romy Lorenz[1,5,6], Mathias Goncalves[1], Eilidh E. MacNicol [7], Christopher J. Markiewicz [1], Yaroslav O. Halchenko[8], Satrajit S. Ghosh [9,10], Krzysztof J. Gorgolewski [1], Russell A. Poldrack [1] & Oscar Esteban [1,11] ✉

Reference anatomies of the brain ('templates') and corresponding atlases are the foundation for reporting standardized neuroimaging results. Currently, there is no registry of templates and atlases; therefore, the redistribution of these resources occurs either bundled within existing software or in ad hoc ways such as downloads from institutional sites and general-purpose data repositories. We introduce TemplateFlow as a publicly available framework for human and non-human brain models. The framework combines an open database with software for access, management, and vetting, allowing scientists to share their resources under FAIR−findable, accessible, interoperable, and reusable−principles. TemplateFlow enables multifaceted insights into brains across species, and supports multiverse analyses testing whether results generalize across standard references, scales, and in the long term, species.

The morphological diversity of brains necessitates a framework for formalizing population-level knowledge about anatomy and function. Neuroscientists have answered this need by creating brain 'templates'−population-average representations of a particular brain imaging modality, a specific cohort, and/or study sample−and brain 'atlases'−annotations that associate spatial locations such as voxels or surface mesh vertices in templates with labels. Together, the development of templates and atlases has accelerated the discovery and dissemination of new knowledge in neuroscience. A substantial number of templates and atlases are now available for different populations, species, image modalities, and coordinate systems. Because they are indispensable to analytic workflows, the most widely used templates and atlases are typically distributed along with neuroimaging software libraries. However, challenges arising from the management, stewardship, distribution, reuse, and reporting of template resources limit the integration of many templates into scientific workflows. Perhaps for these reasons, researchers seldom deviate from using the default template of their software of choice[1].

This software-bound distribution mode has set limits to transparent reporting. Indeed, it is widely assumed that the provenance of templates is completely specified by the choice of software library[1]. We illustrate the limitations of this de facto standard practice vis-à-vis the 'canonical' human neuroimaging template, 'MNI space' (Montreal Neurological Institute), and its distribution within three widely used neuroimaging software libraries: SPM, FSL, and FreeSurfer. Studies carried out with SPM96 and earlier versions report their results in MNI space with reference to the single-subject 'Colin 27' average template[2]. However, updates to SPM99 and subsequent versions alternatively use MNI space to refer to linear[3] or non-linear[4] averages of 152 subjects. By contrast, the MNI template bundled with FSL[5] is not in fact part of the

[1]Department of Psychology, Stanford University, Stanford, CA, USA. [2]Department of Bioengineering, Stanford University, Stanford, CA, USA. [3]Department of Applied Information Technology, University of Gothenburg, Gothenburg, Sweden. [4]Department of Clinical Neuroscience, Karolinska Institutet, Stockholm, Sweden. [5]MRC CBU, University of Cambridge, Cambridge, UK. [6]Department of Neurophysics, MPI, Leipzig, Germany. [7]Department of Neuroimaging, Institute of Psychiatry, Psychology and Neuroscience, King's College London, London, UK. [8]Department of Psychological and Brain Sciences, Dartmouth College, Hanover, NH, USA. [9]McGovern Institute for Brain Research, Massachusetts Institute of Technology, Cambridge, MA, USA. [10]Department of Otolaryngology, Harvard Medical School, Boston, MA, USA. [11]Department of Radiology, University Hospital of Lausanne and University of Lausanne, Lausanne, Switzerland. ✉e-mail: rastko@stanford.edu; phd@oscaresteban.es

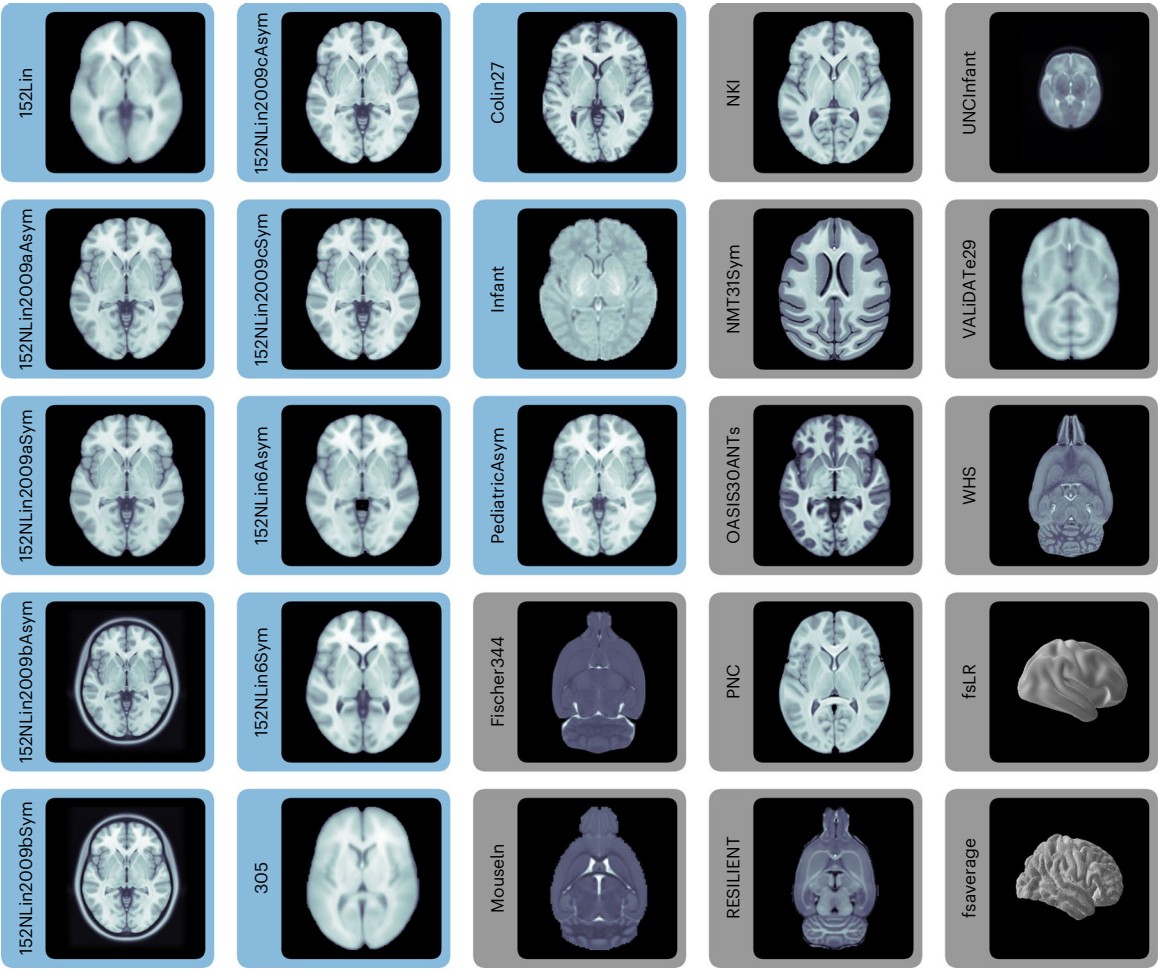

**Fig. 1 | Representative views of 25 templates currently available in the TemplateFlow Archive, including 13 templates from the MNI portfolio (blue).** WHS, Fischer344, RESILIENT, and MouseIn are rodent templates. fsaverage and fsLR are surface templates; the remaining templates are volumetric. Each template is distributed with atlas labels, segmentations, and metadata files.

official portfolio distributed by the MNI. FreeSurfer uses yet another variation of the linear MNI average that is based on 305 subjects[6]. Thus, the single term MNI is used as an ambiguous reference to different data objects. We explore these ambiguities within the topic analysis described in Supplementary Text 1, and illustrate these results in Supplementary Figure 1. Issues regarding access, licensing, unequivocal identification, and integration within software workflows are further exacerbated when a template or atlas is not a software default.

Further limitations emerge when using templates that are not phenotypically proximal to a study's sample. Using inappropriate templates can introduce 'template effects' that bias morphometry and produce incorrect results[7]. Because of the scarcity of templates in non-human imaging and their scant accommodation in software libraries[8], exposure to template effects is even more pressing in this context. For instance, it is unclear the extent to which and the range of applications where it would be appropriate to use a mouse template for the standardization of rat images.

Following the 'Findability, Accessibility, Interoperability, and Reusability (FAIR) Guiding Principles'[9] (Supplementary Note 1), we introduce TemplateFlow to address these challenges. TemplateFlow decouples standardized spatial data from software libraries while affording data workflows[10–12] with the flexibility necessary to select the most appropriate template available. TemplateFlow comprises a cloud-based repository of human and non-human imaging templates—the TemplateFlow Archive (Fig. 1)—paired with a Python-based library—the TemplateFlow Client—for programmatically accessing

template resources. Finally, the TemplateFlow Manager is used to upload new or update existing resources. These software components, as well as all template resources, are version controlled. Thus, not only does TemplateFlow enable off-the-shelf access to templates by humans and machines, it also permits researchers to share their resources with the community (Fig. 2).

The TemplateFlow Archive (Fig. 1) contains a curated and extensible collection of resources (Supplementary Figure 2), comprising primate and rodent templates and atlases. Human templates span the course of postnatal development and include both volumetric images associated with stereotaxic coordinate systems and surface meshes reconstructed to follow cortical geometry. Although most templates currently represent magnetic resonance imaging (MRI)-derived features, TemplateFlow is not limited to any specific modality. An enumeration of currently archived templates is presented in Supplementary Table 1. Cloud storage for the *Archive* is supported by the Open Science Framework (OSF) and Amazon's Simple Storage Service (S3). The online documentation hub and the resource browser located at https://www.templateflow.org/ provide further details for users (Supplementary Figure 2).

To satisfy interoperability and reusability principles, *Template-Flow* defines a vocabulary for unambiguous and machine-readable representation of knowledge and metadata inspired by the Brain Imaging Data Structure (BIDS)[13]. BIDS prescribes a file-naming scheme comprising a series of key–value pairs (called 'entities'). TemplateFlow adopts this design pattern, pairing each template with an entity that

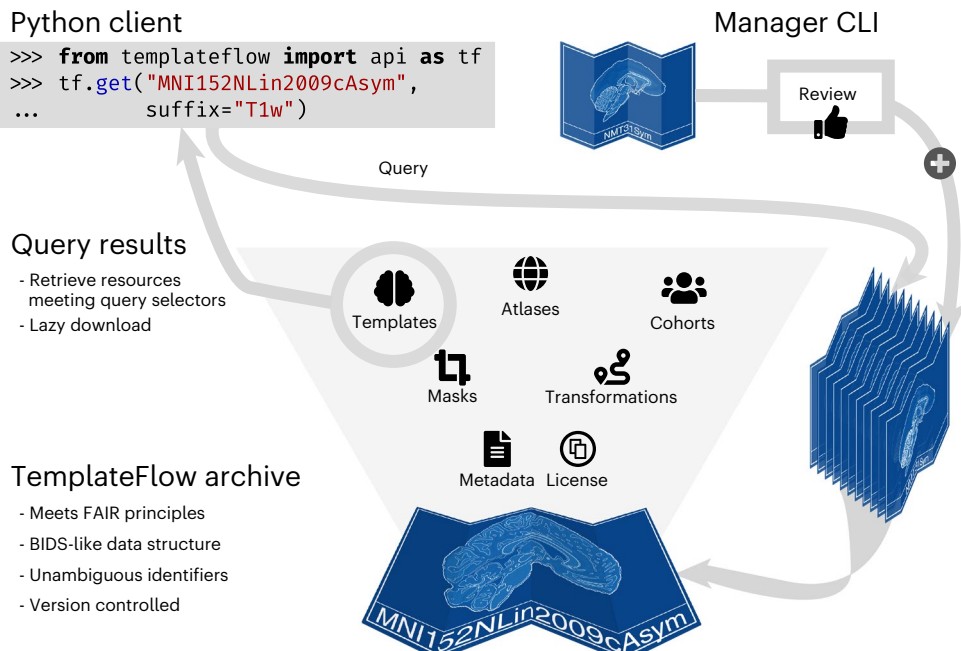

**Fig. 2 | TemplateFlow implements the FAIR guiding principles.** The TemplateFlow Archive can be accessed at a 'low' level with DataLad, or at a 'high' level with a Python client. New resources can be added through the Manager command-line interface, which initiates a peer-review process before acceptance in the Archive.

uniquely and persistently identifies it. This entity is signified with the key 'tpl-' and accepts an alphanumeric string as its value (for example, 'tpl-MNI152Lin'). The unique identifier, together with the version control intrinsic to DataLad[14] data management, resolves the issue of ambiguous reporting (Supplementary Text 2). The BIDS-inspired vocabulary also formalizes the data types and properties of resources associated with templates, such as multi-resolution template images, volumetric or surface-based atlas annotations, geometry files, and label metadata (Supplementary Table 2). The only requirement for feature averages and atlases sharing a unique 'tpl' identifier is that they must be spatially in register. Template resources are described with rich metadata (Supplementary Figure 3) and distributed with a copy of the usage license stipulated by their authors.

The Python client for TemplateFlow provides human users and software tools with a standardized and intuitive protocol for programmatic access to the Archive, as illustrated in Supplementary Figure 4 and Supplementary Note 2. To query TemplateFlow, a user can submit a list of arguments corresponding to the BIDS-like key–value pairs in the file name of each entity (Supplementary Table 2). The client leverages PyBIDS[15] to match user queries against available resources and DataLad[14] to synchronize matched resources from the cloud to local filesystems. The client implements lazy loading: instead of distributing data with the installation, TemplateFlow allows the user to dynamically pull from cloud-based storage only those resources they need, as they need them. After a resource has been requested once, it remains cached in the filesystem for future use.

In addition to providing a platform for accessing the most widely used neuroimaging templates, TemplateFlow is envisioned as a centralized and standard framework for researchers to disseminate new templates. A submission pipeline for templates is streamlined with the Python-based TemplateFlow Manager software, which facilitates resource submission with minimal technical overhead. When adding a new template, the Manager initiates a GitHub-based, public peer-review pipeline where experts are invited to curate and vet new proposals (Supplementary Figure 5). In addition to hosting peer review of new template resources, the TemplateFlow organization on GitHub leverages uses the 'issues' feature of each project as a channel

for community members to propose new template or workflow features. The 'pull-requests' feature enables the direct contribution of code or template resources.

The TemplateFlow management infrastructure reduces the burden of maintenance and promotes early identification of errors. The absence of existing infrastructure for template distribution has given rise to static development modes where templates are packaged once and rarely revisited, remaining outside of version control. Nevertheless, errors in existing template resources have been reported (for example, https://www.jiscmail.ac.uk/cgi-bin/webadmin?A2=fsl;e46ecc4c.1210, accessed January 2022) and documented[16]. In our experience with OpenfMRI and OpenNeuro[17], maximizing reuse leads to early error identification and subsequent dataset revision.

Without a centralized repository designed around FAIR principles, deviating from software default templates at present requires knowing how to find and access template resources and how to integrate them into a workflow. Among established software libraries, only AFNI[18] has, to our knowledge, taken steps to mitigate this burden on users with the '@Install_<template_name>' functionality for retrieving template resources, although this utility: (i) places the maintenance burden and costs on the AFNI team; (ii) does not expose versioning of templates to users; (iii) is not designed to allow researchers to spontaneously upload their templates; and (iv) offers only straightforward download through its client. Supplementary Table 4 showcases how TemplateFlow covers the gaps specific to templates and atlases and their reuse, in comparison to the general purpose 'NeuroImaging Tools & Resources Collaboratory' (NITRC).

We demonstrate benefits of centralizing templates via the integration of TemplateFlow into fMRIPrep[12], a functional MRI preprocessing tool. This integration provides fMRIPrep users with flexibility to spatially normalize their data to any template available in the Archive. This integration has accelerated the adaptation of fMRIPrep to pediatric populations[19] and rodent imaging[20], using suitable templates from the Archive. The uniform interface provided by the BIDS-like directory organization and metadata enables straightforward integration of new templates into workflows equipped to use TemplateFlow templates (Supplementary Note 3).

One point of concern is that TemplateFlow affords researchers substantial analytical flexibility in the choice of standard spaces of reference. Empirical investigations into the consequences of flexibility in neuroimaging[21,22] have demonstrated that decision points in workflows can lead to substantial variability in analysis outcomes, sometimes manifesting as irreproducible results. Contemporaneous work[1] underscored that analytical variability degrades the reproducibility of studies only in combination with (intended or unintended) selective reporting of methods and results. Selective reporting, in this particular application, would mean that a researcher explores the results with reference to several templates or atlases but fails to comprehensively report all the experimental options traversed. Using DataLad or the TemplateFlow Client, researchers have at their disposal the necessary tooling for precise reporting: unique identifiers, provenance tracking, version tags, and comprehensive metadata. Therefore, the effectiveness of TemplateFlow to mitigate selective reporting is bounded by the user's discretion.

In addition to promoting exact reporting, Botvinik-Nezer et al.[22] advocated further exploring a 'multiverse' of analyses, wherein many combinations of methodological choices are all thoroughly reported and cross-compared when presenting results. A FAIR resource like TemplateFlow answers a critical need to aid researchers in exploring multiple template and atlas combinations with ease, to ultimately determine whether their research findings hold robustly across the multiverse of workflow configurations. The TemplateFlow client empowers users to incorporate multiverse analysis into their research by easily making template or atlas substitutions for cross comparison.

Leveraging our experience developing OpenNeuro[17] and related research instruments[11,12], we introduce an open framework for the archiving, maintenance and sharing of neuroimaging templates and atlases called TemplateFlow that is implemented under FAIR data sharing principles. The Supplementary Text 2 linked to this paper expands on the current need for this resource in the domain of neuroimaging, and further discusses the implications of the increased analytical flexibility this tool affords (Supplementary Discussion). The approach taken by TemplateFlow to addressing both the availability of resources under FAIR principles and analytical flexibility establishes a pattern broadly transferable beyond neuroimaging. We envision TemplateFlow as a core research tool undergirding multiverse analyses—assessing whether neuroimaging results are robust across population-wide spatial references—as well as a stepping stone towards the quest of mapping anatomy and function across species.

## Online content

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

## Methods

TemplateFlow comprises three cardinal components: (i) a cloud-based archive; (ii) a Python client for programmatically querying the archive; and (iii) automated systems for synchronizing and updating archive data.

### The TemplateFlow Archive

The archive itself comprises directories of template data in cloud storage. For redundancy, the data are stored on both Google Cloud using the OSF and Amazon S3. Before storage, all template data must be named and organized in directories conforming to a data structure adapted from the BIDS standard[13]. The precise implementation of this data structure is a living document and is detailed on the TemplateFlow homepage (https://www.templateflow.org).

The archive is organized hierarchically, and descriptive metadata follow a principle of inheritance: any metadata that apply to a particular level of the archive also apply to all deeper levels. At the top level of the hierarchy are directories corresponding to each archived template. If applicable, within each template directory are directories corresponding to sub-cohort templates. Names of directories and resource files constitute a hierarchically ordered series of key–value pairs terminated by a suffix denoting the datatype. For instance, 'tpl-MNIPediatricAsym_cohort-3_res-high_T1w.nii.gz' denotes a $T_1$-weighted template image file for resolution 'high' of cohort '3' in the 'MNIPediatricAsym' template (where the definitions of each resolution and cohort are specified in the template metadata file). The most common TemplateFlow data types are indexed in Fig. 2 and Supplementary Table 2; an exhaustive list is available in the most current version of the BIDS standard (https://bids-specification.readthedocs.io/).

Within each directory, template resources include image data, atlas and template metadata, transform files, licenses, and curation scripts. All image data are stored in gzipped NIfTI-1 format and are conformed to 'RAS+' orientation (that is, left-to-right, posterior-to-anterior, inferior-to-superior, with the affine 'q-form' and 's-form' matrices corresponding to a cardinal basis scaled to the resolution of the image). Template metadata are stored in a JavaScript Object Notation file called 'template_description.json'; an overview of metadata specifications is provided in Supplementary Figure 3. In brief, template metadata files contain general template metadata (for example, authors and curators, references), cohort-specific metadata (for example, ages of subjects included in each cohort), and resolution-specific metadata (for example, dimensions of images associated with each resolution). Atlas metadata are often stored in tab-separated values format and specify the region name corresponding to each atlas label. Template-to-template transform files are stored in HDF5 format and are generated as a diffeomorphic composition of Insight Toolkit-formatted transforms mapping between each pair of templates.

The archive has a number of client-facing access points to facilitate browsing of resources. Key among these is the archive browser on the TemplateFlow homepage, which indexes all archived resources and provides a means for researchers to take inventory of possible templates to use for their study.

### The Python client

TemplateFlow is distributed with a Python client that can submit queries to the archive and download any resources as they are requested by a user or program. Valid query options correspond approximately to BIDS key–value pairs and data types. A compendium of common query arguments is provided in Supplementary Table 2, and comprehensive documentation is available on the TemplateFlow homepage (https://www.templateflow.org).

When a query is submitted to the TemplateFlow client, the client begins by identifying any files in the archive that match the query. To do so, it uses PyBIDS[15], which exploits the BIDS-like architecture of the TemplateFlow Archive to efficiently scan all directories and filter any matching files. Next, the client assesses whether queried files exist as data in local storage. When a user locally installs TemplateFlow, the local installation initially contains only lightweight pointers to files in OSF's cloud storage. These pointers are implemented using DataLad[14], a data management tool that extends git and git-annex. TemplateFlow uses DataLad principally to synchronize datasets across machines and to perform version control by tracking updates made to a dataset.

If the queried files are not yet synchronized locally (that is, they exist only as pointers to their counterparts in the cloud), the client instructs DataLad to retrieve them from cloud storage. In the event that DataLad fails or returns an error, the client falls back on redundancy in storage and downloads the file directly from Amazon S3. When the client is next queried for the same file, it will detect that the file has already been cached in the local filesystem. The use of resource pointers with the client thus enables lazy loading of template resources. Finally, the client confirms that the file has been downloaded successfully. If the client detects a successful download, it returns the result of the query; in the event that it detects a synchronization failure, it displays a warning for each queried file that encountered a failure.

Continued functionality and operability of the client is ensured through an emphasis on maximizing code coverage with unit tests. Updating the client requires successful completion of all unit tests, which are automatically executed by continuous integration and continuous delivery services connected to GitHub. Continuous integration and continuous delivery also keep the web-based archive browser up to date by automatically indexing data files.

### Ancillary and managerial systems

TemplateFlow includes a number of additional systems and programs that serve to automate stages of the archive update process, for instance addition of a new template or revision of current template resources. To facilitate the update and extension process, TemplateFlow uses GitHub 'actions' to automatically synchronize dataset information so that all references remain up to date with the current dataset. These actions are triggered whenever a pull request to TemplateFlow is accepted. For example, GitHub actions are used to update the browser of the TemplateFlow Archive so that it displays all template resources as they are uploaded to the archive.

Whereas the TemplateFlow Client synchronizes data from cloud storage to the local filesystem, a complementary TemplateFlow Manager handles the automated synchronization of data from the local filesystem to cloud storage. The Python-based manager is also used for template intake, that is, to propose the addition of new templates to the archive. To propose adding a new template, a user first runs the TemplateFlow Manager using the 'tfmgr add <template_id> --osf-project <project_id>' command. Additional details of the command-line interface for the TemplateFlow Manager are indexed in Supplementary Table 3.

The manager begins by using the TemplateFlow client to query the archive and verify that the proposed template does not already exist. After verifying that the proposed template is new, the manager synchronizes all specified template resources to OSF's cloud storage. It then creates a fork of the 'tpl-intake' branch of the 'templateflow' GitHub repository and generates an intake file in Tom's Obvious Minimal Language markup format; this intake file contains a reference to the OSF project where the manager has stored template resources. The TemplateFlow Manager commits the Tom's Obvious Minimal Language intake file to the fork and pushes to the user's GitHub account. Finally, it retrieves template metadata from 'template_description.json' and uses the metadata to compose a pull request on the 'tpl-intake' branch. This pull-request provides a venue for discussion and vetting of the proposed addition of a new template.

### Ethical compliance

We complied with all relevant ethical regulations. This resource reused publicly available data derived from studies acquired at many different

institutions. Protocols for all of the original studies were approved by the corresponding ethical boards.

## Reporting summary

Further information on research design is available in the Nature Portfolio Reporting Summary linked to this article.

## Data availability

All templates and associated data are available from https://www.templateflow.org under corresponding open licenses and accessible as described in the manuscript.

## Code availability

All the software components discussed in this paper are available under the Apache 2.0 license and accessible from https://github.com/templateflow. The resource is demonstrated in a Code Capsule linked to this article (https://doi.org/10.24433/CO.1580121.v1).

## Acknowledgements

The development of this resource was supported by the Laura and John Arnold Foundation (R.A.P. and K.J.G.), the National Institute of Biomedical Imaging and Bioengineering (R01EB020740, S.S.G.; P41EB019936 S.S.G., Y.O.H.), National Institute of Mental Health (RF1MH121867 R.A.P., O.E.; R24MH114705 and R24MH117179, R.A.P.; 1RF1MH121885 S.S.G.), National Institute of Neurological Disorders and Stroke (U01NS103780, R.A.P.), and National Science Foundation (CRCNS 1912266, Y.O.H.). R.L. is funded by the Wellcome Trust (209139/Z/17/Z). E.M. was supported by the UK Medical Research Council (MR/N013700/1) and King's College London. O.E. acknowledges financial support from the Swiss National Science Foundation Ambizione project "Uncovering the interplay of structure, function, and dynamics of brain connectivity using MRI" (grant number 185872).

## Author contributions

Conceptualization: R.C., C.J.M., K.J.G., R.A.P., O.E.; data curation: R.C., E.M., O.E.; topics analysis: R.C., R.L., O.E.; funding acquisition: K.J.G., R.A.P., O.E.; methodology: R.C., O.E.; project administration: R.A.P., O.E.; resources: Y.O.H., S.S.G., K.J.G., R.A.P., O.E.; software and documentation: R.C., W.H.T., M.G., E.M., C.J.M., Y.O.H., O.E.; supervision: R.A.P., O.E.; validation: R.C., W.H.T., E.M., O.E.; visualization: R.C., R.L.; writing (original draft): R.C., O.E.; writing (review and editing): R.C., R.L., W.H.T., M.G., E.M., C.J.M., Y.O.H., S.S.G., K.J.G., R.A.P., O.E.

## Competing interests

The authors declare no competing interests.

## Additional information

**Correspondence and requests for materials** should be addressed to Rastko Ciric or Oscar Esteban.

Rastko Ciric <rastko@stanford.edu>

# Reporting Summary

## Statistics

For all statistical analyses, confirm that the following items are present in the figure legend, table legend, main text, or Methods section.

| n/a | Confirmed | |
|---|---|---|
| ☒ | ☐ | The exact sample size (*n*) for each experimental group/condition, given as a discrete number and unit of measurement |
| ☒ | ☐ | A statement on whether measurements were taken from distinct samples or whether the same sample was measured repeatedly |
| ☒ | ☐ | The statistical test(s) used AND whether they are one- or two-sided *Only common tests should be described solely by name; describe more complex techniques in the Methods section.* |
| ☒ | ☐ | A description of all covariates tested |
| ☒ | ☐ | A description of any assumptions or corrections, such as tests of normality and adjustment for multiple comparisons |
| ☒ | ☐ | A full description of the statistical parameters including central tendency (e.g. means) or other basic estimates (e.g. regression coefficient) AND variation (e.g. standard deviation) or associated estimates of uncertainty (e.g. confidence intervals) |
| ☒ | ☐ | For null hypothesis testing, the test statistic (e.g. *F*, *t*, *r*) with confidence intervals, effect sizes, degrees of freedom and *P* value noted *Give P values as exact values whenever suitable.* |
| ☒ | ☐ | For Bayesian analysis, information on the choice of priors and Markov chain Monte Carlo settings |
| ☒ | ☐ | For hierarchical and complex designs, identification of the appropriate level for tests and full reporting of outcomes |
| ☒ | ☐ | Estimates of effect sizes (e.g. Cohen's *d*, Pearson's *r*), indicating how they were calculated |

*Our web collection on statistics for biologists contains articles on many of the points above.*

## Software and code

Policy information about availability of computer code

Data collection: This is a resource for the redistribution under FAIR principles of brain models derived from neuroimaging data and therefore does not have a corresponding data collection procedure.
However, all the data made available by TemplateFlow must be licensed under permissive, open terms prior to the inclusion within the resource.
In addition, the three main components of TemplateFlow are distributed under permissive licenses:
- The Archive: https://github.com/templateflow/templateflow
- The Python client: https://github.com/templateflow/python-client
- The Manager: https://github.com/templateflow/python-manager

Data analysis: Three jupyter notebooks illustrate the utilization of TemplateFlow and have been used to generate Figure 1 and Supplementary Table 1 of the manuscript: https://github.com/templateflow/templateflow-notebooks

For manuscripts utilizing custom algorithms or software that are central to the research but not yet described in published literature, software must be made available to editors and reviewers. We strongly encourage code deposition in a community repository (e.g. GitHub). See the Nature Portfolio guidelines for submitting code & software for further information.

## Data

Policy information about availability of data

All manuscripts must include a data availability statement. This statement should provide the following information, where applicable:

- Accession codes, unique identifiers, or web links for publicly available datasets
- A description of any restrictions on data availability
- For clinical datasets or third party data, please ensure that the statement adheres to our policy

> All data made available by TemplateFlow must be under permissive licenses (e.g., CC0, CC-BY, etc.)

## Human research participants

Policy information about studies involving human research participants and Sex and Gender in Research.

| | |
|---|---|
| Reporting on sex and gender | n/a |
| Population characteristics | n/a |
| Recruitment | n/a |
| Ethics oversight | n/a |

Note that full information on the approval of the study protocol must also be provided in the manuscript.

# Field-specific reporting

Please select the one below that is the best fit for your research. If you are not sure, read the appropriate sections before making your selection.

☒ Life sciences ☐ Behavioural & social sciences ☐ Ecological, evolutionary & environmental sciences

For a reference copy of the document with all sections, see nature.com/documents/nr-reporting-summary-flat.pdf

# Life sciences study design

All studies must disclose on these points even when the disclosure is negative.

| | |
|---|---|
| Sample size | n/a |
| Data exclusions | n/a |
| Replication | n/a |
| Randomization | n/a |
| Blinding | n/a |

# Reporting for specific materials, systems and methods

We require information from authors about some types of materials, experimental systems and methods used in many studies. Here, indicate whether each material, system or method listed is relevant to your study. If you are not sure if a list item applies to your research, read the appropriate section before selecting a response.

## Materials & experimental systems

| n/a | Involved in the study |
|-----|----------------------|
| ☒ ☐ | Antibodies |
| ☒ ☐ | Eukaryotic cell lines |
| ☒ ☐ | Palaeontology and archaeology |
| ☒ ☐ | Animals and other organisms |
| ☒ ☐ | Clinical data |
| ☒ ☐ | Dual use research of concern |

## Methods

| n/a | Involved in the study |
|-----|----------------------|
| ☒ ☐ | ChIP-seq |
| ☒ ☐ | Flow cytometry |
| ☒ ☐ | MRI-based neuroimaging |

