## [Peer Review File · Nature Methods]

Peer Review Information

Manuscript Title: TemplateFlow: FAIR-sharing of multi-scale, multi-species brain models

Corresponding author name(s): Oscar Esteban, Rastko Ciric

Editorial Notes:

Redactions – published data Parts of this Peer Review File have been redacted as indicated to remove third-party material.

Reviewer Comments & Decisions:

Decision Letter, initial version:

Dear Oscar,

Thank you for your patience. Your Resource, "TemplateFlow: FAIR-sharing of multi-scale, multi-species brain models", has now been seen by three reviewers. As you will see from their comments below, although the reviewers find your work of potential interest, they have raised a number of concerns. We are interested in the possibility of publishing your paper in Nature Methods, but would like to consider your response to these concerns before we reach a final decision on publication.

We therefore invite you to revise your manuscript to address these concerns. Importantly, please do make sure to clearly state benefits over the resources provided at NITRC.

In addition, we ask that you revise the manuscript to fit our Brief Communications format (2 figures, 1500 words).

[Redacted]

This URL links to your confidential home page and associated information about manuscripts you may have submitted, or that you are reviewing for us. If you wish to forward this email to co-authors, please delete the link to your homepage.

We hope to receive your revised paper within about 6 weeks. If you cannot send it within this time, please let us know. In this event, we will still be happy to reconsider your paper at a later date so long as nothing similar has been accepted for publication at Nature Methods or published elsewhere.

OPEN SCIENCE REQUIREMENTS

REPORTING SUMMARY AND EDITORIAL POLICY CHECKLISTS

Please note that these forms are dynamic 'smart pdfs' and must therefore be downloaded and completed in Adobe Reader. We will then flatten them for ease of use by the reviewers. If you would

like to reference the guidance text as you complete the template, please access these flattened versions at <http://www.nature.com/authors/policies/availability.html>.

Please make sure to update the Code Ocean compute capsule if you are revising the code.

DATA AVAILABILITY

All novel DNA and RNA sequencing data, protein sequences, genetic polymorphisms, linked genotype and phenotype data, gene expression data, macromolecular structures, and proteomics data must be deposited in a publicly accessible database, and accession codes and associated hyperlinks must be provided in the “Data Availability” section.

Please include a “Data availability” subsection in the Online Methods. This section should inform readers about the availability of the data used to support the conclusions of your study, including accession codes to public repositories, references to source data that may be published alongside the paper, unique identifiers such as URLs to data repository entries, or data set DOIs, and any other statement

about data availability. At a minimum, you should include the following statement: “The data that support the findings of this study are available from the corresponding author upon request”, describing which data is available upon request and mentioning any restrictions on availability. If DOIs are provided, please include these in the Reference list (authors, title, publisher (repository name), identifier, year). For more guidance on how to write this section please see: <http://www.nature.com/authors/policies/data/data-availability-statements-data-citations.pdf>

CODE AVAILABILITY

Please include a “Code Availability” subsection in the Online Methods which details how your custom code is made available. Only in rare cases (where code is not central to the main conclusions of the paper) is the statement “available upon request” allowed (and reasons should be specified).

MATERIALS AVAILABILITY

ORCID

Nature Methods is committed to improving transparency in authorship. As part of our efforts in this direction, we are now requesting that all authors identified as ‘corresponding author’ on published papers create and link their Open Researcher and Contributor Identifier (ORCID) with their account on

the Manuscript Tracking System (MTS), prior to acceptance. This applies to primary research papers only. ORCID helps the scientific community achieve unambiguous attribution of all scholarly contributions. You can create and link your ORCID from the home page of the MTS by clicking on 'Modify my Springer Nature account'. For more information please visit www.springernature.com/orcid.

Best regards,
Nina

Nina Vogt, PhD
Senior Editor
Nature Methods

Reviewers' Comments:

Reviewer #1:

Remarks to the Author:

A. Summary of the key results

The main deliverable here is a toolbox that allows for better management and use of brain templates, with the specific criterion of conforming to the FAIR principles. TemplateFlow makes templates:

- Findable: a unique identifier is associated with each template (and metadata, atlas etc.).
- Accessible: data are accessible using open, free, standard communications protocols without need for authentication (the DataLad approach is used)
- Interoperable: A BIDS-type structure is applied to the templates.
- Reusable: license usage, provenance and standards are based on the BIDS format.

B. Originality and significance: if not novel, please include reference

The brain templates are already available. This manuscript does not address any methods of developing and evaluating templates. This toolbox does address the important issue of imprecise description of

templates in manuscripts (e.g., papers often describe the ‘MNI template’ but there are several versions that fall under this rubric) by adapting existing standards and methods (e.g., BIDS and datalad). The problem in the field is that templates are typically embedded within particular software packages and users tend to stay with the default settings. This may not be a huge concern if a study is entirely self-contained and an adult template is used for adult subjects, for example. However, it is particularly important when considering the sharing of raw data – results using one version of a template (e.g., SPM’s ‘MNI’ template) are not necessarily directly comparable to another (e.g., FSL’s ‘MNI’ template). In theory, users can obtain, curate and document use of these templates without using TemplateFlow. The novelty of this toolbox is it provides easier access to a range of templates and to better document the specific templates used.

The significance of this method is that reproducibility of MRI studies will be improved with better documentation of template use.

C. Data & methodology: validity of approach, quality of data, quality of presentation

Validity of approach: the toolbox is implemented through Python, loading templates (as requested) from cloud storage. The core aspects of Template flow are:

The “Archive”: a repository of human and non-human templates. There are currently 15 templates available, primarily volumetric: human adult and pediatric brains and rat brains. More templates can be added. The structure of the Archive follows the BIDS format.

The “Client” is a Python-based library for programmatically accessing templates in the Archive. Key-value pair arguments can be used to select specific templates.

The “Manager” tool facilitates the addition of new templates/resources. User files stored locally are uploaded to cloud storage and GitHub repository is created that allows pull requests.

-Comment: is not clear to me if new templates generated by users are immediately available in the archive: In 128: “When adding a new template, the Manager initiates a peer-reviewed contribution pipeline where experts are invited to curate and vet new proposals” versus In 220: “Synchronization of spatial data assets to the TemplateFlow Archive affords data producers an immediate way to distribute their data according to FAIR principles”

I have run the code provided, and it was very intuitive and well annotated.

The figures in the manuscript were useful.

D. Appropriate use of statistics and treatment of uncertainties

This manuscript primarily described the toolbox. The implementation of the topic model analysis (LDA with learning decay = .7) seemed reasonable.

E. Conclusions: robustness, validity, reliability

The toolbox is implemented in Python, often utilizing extant software and standards. It is therefore likely

quite robust.

Validity – The templates in the Archive are all publicly available and have been subjected to scrutiny by the neuroimaging community. With respect to new templates, a peer-review process ensures that all data are conformant with the TemplateFlow standard, but presumably it is possible that faulty/deficient templates could be uploaded. However, the faulty templates would be easy to trace.

Reliability – the implementation of version control and use of Research Resource Identifiers will aid reliability. The templates can be integrated into workflows and the code-based nature. It seems unlikely that users could inadvertently use or describe the wrong template.

F. Suggested improvements: experiments, data for possible revision

Ln 675: TemplateFlow can allow users to calculate ‘robust’ transformations between any pair of adult human template spaces: (easy) inter-template registration is very useful. The registration uses 10 images from MNI-152 via ANTs. This is described very briefly in the manuscript – more information would be helpful, especially for those who have not used ANTs before.

Currently, the Archive consists of MRI templates, but could potentially include other modalities. It would be very interesting to see some other modalities and this would increase the scope of the manuscript. TemplateFlow uses lazy loading. After a resource has been requested once, it is then cached. Although users are probably not going to access so many templates that storage is an issue but: can the cache be cleared?

The topic model analysis is interesting but feels like it could be a starting place, not at the end of the Results. Perhaps, it could be moved to Supplemental and mentioned in the Introduction as evidence for the tendency for template choice to be a function of software defaults?

G. References: appropriate credit to previous work?

Yes, previous work and software are well referenced.

H. Clarity and context: lucidity of abstract/summary, appropriateness of abstract, introduction and conclusions

The Abstract, Methods, Results and Discussion are generally quite clear. The motivation for the toolbox is described well. Some minor comments.

The Introduction could be better structured. There are some long paragraphs that could be split up (e.g., start new paragraph on line 36 ‘On account...’)

Results (Ln 137).

‘FAIR’ is in the title of the manuscript but the FAIR principles themselves are only listed in a supplemental box, they could be moved to the main text. The results could also work though FAIR more systematically (one paragraph per principle?).

In the Discussion, the authors suggest that one limitation of TemplateFlow is increased potential for

methodological flexibility. But this is not a limitation of the toolbox. Presumably, users could now preregister the exact template that they plan to use?

Reviewer #2:

Remarks to the Author:

The current work provides general background on the use of anatomical MRI atlases in neuroimaging research, as well as some of the common issues surrounding misreporting and replication difficulties that arise due to differences in "default" atlases provided by popular neuroimaging software. The tool presented in this paper, TemplateFlow, seeks to mitigate these issues by providing a database of common atlases, a python "client" for the integration of TemplateFlow into preprocessing pipelines, thorough atlas file metadata for precise reporting and ease of replication, and a "manager" for users to upload new atlases which are then peer reviewed before being added to the database.

This manuscript is generally well written, clearly outlines the need for a tool such as TemplateFlow, and is accompanied by in-text figures with example code and jupyter notebooks that are available online. The provided examples clearly walk potential users through the basics of using TemplateFlow in a way that is easy to follow and understand, given the user has a very basic knowledge of python. While seasoned fMRI researchers are used to downloading different atlases and hunting down the technical specifics in supplemental materials for use and manuscript reporting, TemplateFlow can greatly mitigate mistakes, misreporting, and also provides clear information for trainees with limited background in this area. I feel that tools like TemplateFlow and manuscripts such as this that outline methodological pitfalls and provide a clear and valid solution are vital for trainees and the development of robust replication practices in our field. I enthusiastically support this work being accepted for publication after the authors address the minor issues outlined below.

Minor Issues:

The jupyter notebook that is hosted online worked flawlessly, was clear, and gave a nice walkthrough of what the different TemplateFlow functions were doing and how they could be integrated into a preprocessing pipeline. However, when working through some of the practical examples in the manuscript, there were some typos or errors that I feel should be corrected or updated for clarity or the benefit of less-seasoned readers.

1. I tested the installation and use of the python package locally and the majority of tests were without error. However, I could not follow the example in Figure 5. The get command would return an empty value when following the example that was supposed to pull an atlas while providing filter specifics

about the template (desc, resolution, etc). I then slowly tested and removed fields. This function only pulled files and returned a non-empty variable when I provided only the identifier/name, but then it started pulling all MNI152Nlin6Asym niftis (quite a few) and I cancelled the downloads.

I tested this a second time, and this time I was confused whether it was returning an empty variable because I had just previously started downloading all MNI152Nlin6Asym files and the file already existed locally, or if the example code, in fact did not work. If this issue that I encountered while following the text in figure 5 is an error, I would request it is updated so this step works and is easy to follow and replicate the outputs shown in the figure. If this was a case of user error, I would guess that a new trainee would be confused as well, and some clear explanation of why following the example would result in what I experienced should be included.

2. Supp BoxS2 (minor edit)

In the 4th example:

```
>>> print(get_citations("UNCInfant", bibtex=True)[0])
```

This should have:

```
from templateflow.api import get_citations
```

at the top. I was able to assume this and follow along (even with the missing dependency error that I assume is the reason for the warning that the output is for demonstration only). However, for a new trainee this paper could be a great learning source if it was easy to follow along with all examples without errors. (even if outputs aren't correct or fully implemented yet). This is a very minor edit, but one that I would have appreciated when I was an RA and first learning python.

Reviewer #3:

Remarks to the Author:

This paper describes a resource for archiving, maintaining, and disseminating templates and atlases for the neuroimaging community. To this end, the authors define a series of rules, meta structures, and data structures to systematically and uniformly organize these templates/atlas. Although this resource could provide a new way to organize, document, and disseminate popular templates/atlas, it is unlikely to create broad and significant impact especially in view of existing and popular resources such as NIH-supported NeuroImaging Tools and Resources Collabotary (NITRC, <https://www.nitrc.org>). NITRC is equipped with cloud-based computational and storage resources and covers, among many other things, atlases/templates and computational tools. Taken together with the issues listed below, this work, while not without merits, is not sufficient for Nature Methods. A revision is not likely to change

this outcome.

1. The motivation for this work as claimed by the authors is significant. However, this work does not solve the more pressing issue of methodological variability across studies. Unlike imaging data themselves, which can be large in terms of scale, there are just a handful of atlases and building an elaborate resource like TemplateFlow might be an overkill. The future prospect of TemplateFlow is also not clear, since the number of templates is not expected to grow by much, unlike the imaging data themselves.
2. Grant information should be included in the metadata of each template resource.
3. It is unclear how TemplateFlow manages variations of templates, e.g., ICBM 2009 (a,b,c) Nonlinear (A)Symmetric. For example, will the variations be grouped under a single template resource?
4. The improvement of the analytical flexibility across disciplines as the authors claimed is not supported. The current results do not give enough insight into how TemplateFlow can improve analytical flexibility.
5. It is unclear how TemplateFlow can keep the templates up to date. This issue becomes increasingly critical as the number of templates grows. There is no incentive for the template creators to upload their templates to TemplateFlow, more so when it comes to future maintenance. Most people would just download the latest templates through official channels provided by the template developers.
6. The authors mentioned integration with fMRIPrep with TemplateFlow. But how will TemplateFlow integrate with other processing pipelines, such as FreeSurfer, FSL, and ANTs, which are more geared towards processing of structural MRI data.
7. Surface templates cannot be found on the TemplateFlow website.
8. "The quality of peer reviewed template resources is assessed once prior to publication ...". Why is this considered a poor resource adoption? The templates submitted to TemplateFlow are not going to be reviewed again to suggest changes. It is not clear how TemplateFlow will actually be conducive to the improvement of the templates.
9. The relevance of topic modeling analysis is unclear.
10. Unclear what benefits TemplateFlow brings in comparison with popular platforms such as NITRC. Will the benefits bring significant impact to the field not already fulfilled by NITRC?

Author Rebuttal to Initial comments

Authors' Response to Reviews of

R. Ciric et al. – “TemplateFlow: FAIR-sharing of multi-scale, multi-species brain models”

AR: Authors' Response

We thank the reviewers for their time and interest, as well as their useful feedback. We have made extensive edits on our manuscript, in an attempt to capture all the improvements suggested. We have also shortened the manuscript (in some 70 lines from Introduction through Conclusion, and 10 additional lines from Online Methods) and focused on better clarity, especially attending to comments from Reviewer #3, who seemed the most critical of the work. When referenced, lines point to the annotated version of the revised manuscript, which also includes highlights of added text and the removed text marked with strike-through lines. The document also includes annotations for major changes corresponding to the specific reviewers' comment(s) that prompted the edit.

In addition to the “diff” version, a document with clean typesetting is also provided including annotations for line numbers only. Both versions are automatically generated from the same LaTeX source (i.e., there should not exist any differences between them in terms of this submission's content).

R#0.1: General comment by editors

Importantly, please do make sure to clearly state benefits over the resources provided at NITRC.

t: We thank the editors for bringing this issue to our attention. As part of the thoughtful revision and responding to this comment and comments 3.1 and 3.15 by R#3, we have clearly stated the benefits over NITRC and resources provided therein as follows:

1. We have added a comparison table (Supplementary Table S5) that clearly shows the widely different purposes of the two resources.
2. We have included a specific paragraph discussing the issue within the manuscript (see ll. 260–273). Particularly, the new passage documents several instances in which NITRC's generality makes it inadequate for the specific purpose of disseminating brain models.

Moreover, we have added *TemplateFlow* to NITRC¹ to illustrate the higher hierarchy of NITRC as a resource registry and how both fit in together in the broader picture.

Our Code Capsule demonstrates simple, yet effective, research use-cases that are heavily burdensome to implement without *TemplateFlow*.

¹<https://www.nitrc.org/projects/templateflow>

1. Reviewer #1

R#1.1: Summary of the key results, Originality and significance

The main deliverable here is a toolbox that allows for better management and use of brain templates, with the specific criterion of conforming to the FAIR principles. TemplateFlow makes templates:

- *Findable: a unique identifier is associated with each template (and metadata, atlas etc.).*
- *Accessible: data are accessible using open, free, standard communications protocols without need for authentication (the DataLad approach is used)*
- *Interoperable: A BIDS-type structure is applied to the templates.*
- *Reusable: license usage, provenance and standards are based on the BIDS format.*

The brain templates are already available. This manuscript does not address any methods of developing and evaluating templates. This toolbox does address the important issue of imprecise description of templates in manuscripts (e.g., papers often describe the 'MNI template' but there are several versions that fall under this rubric) by adapting existing standards and methods (e.g., BIDS and datalad). The problem in the field is that templates are typically embedded within particular software packages and users tend to stay with the default settings. This may not be a huge concern if a study is entirely self-contained and an adult template is used for adult subjects, for example. However, it is particularly important when considering the sharing of raw data – results using one version of a template (e.g., SPM's 'MNI' template) are not necessarily directly comparable to another (e.g., FSL's 'MNI' template). In theory, users can obtain, curate and document use of these templates without using TemplateFlow. The novelty of this toolbox is it provides easier access to a range of templates and to better document the specific templates used. The significance of this method is that reproducibility of MRI studies will be improved with better documentation of template use.

AR: We overall agree with the referee's assessment and appreciate the praising words.

R#1.2: Data and methodology

Validity of approach: the toolbox is implemented through Python, loading templates (as requested) from cloud storage. The core aspects of Template flow are:

The "Archive": a repository of human and non-human templates. There are currently 15 templates available, primarily volumetric: human adult and pediatric brains and rat brains. More templates can be added. The structure of the Archive follows the BIDS format.

The "Client" is a Python-based library for programmatically accessing templates in the Archive. Key-value pair arguments can be used to select specific templates.

The "Manager" tool facilitates the addition of new templates/resources. User files stored locally are uploaded to cloud storage and GitHub repository is created that allows pull requests.

AR: We overall agree with the referee's assessment.

R#1.3: Data and methodology

Comment: is not clear to me if new templates generated by users are immediately available in the archive: In 128: "When adding a new template, the Manager initiates a peer-reviewed contribution pipeline where experts are invited to curate and vet new proposals" versus In 220: "Synchronization of spatial data assets to the TemplateFlow Archive affords data producers an immediate way to distribute their data according to FAIR principles"

AR: We thank the reviewer for pointing out this inconsistency. As the reviewer first guesses, new templates are not directly available in the Archive but are first subject to a peer review process. We have completely re-structured the particular section of results to improve the overall clarity (1.12, and 1.13), deferring the description of the upload process to the Online Methods document. We have also made sure this inconsistency was eliminated.

R#1.4: Data and methodology

I have run the code provided, and it was very intuitive and well annotated. The figures in the manuscript were useful.

AR: We thank the reviewer for evaluating the tutorial code, and appreciate the positive contribution of the code resource to enhancing the overall value of the manuscript.

R#1.5: Appropriate use of statistics and treatment of uncertainties

This manuscript primarily described the toolbox. The implementation of the topic model analysis (LDA with learning decay = .7) seemed reasonable.

AR: Following 1.10 and other reviewers' suggestions, we have moved the topic modeling to the Supplementary Materials. We nonetheless appreciate the assessment by the reviewer, which raises confidence in this aspect of the work.

R#1.6: Conclusions

The toolbox is implemented in Python, often utilizing extant software and standards. It is therefore likely quite robust.

Validity – The templates in the Archive are all publicly available and have been subjected to scrutiny by the neuroimaging community. With respect to new templates, a peer-review process ensures that all data are conformant with the TemplateFlow standard, but presumably it is possible that faulty/deficient templates could be uploaded. However, the faulty templates would be easy to trace.

Reliability – the implementation of version control and use of Research Resource Identifiers will aid reliability. The templates can be integrated into workflows and the code-based nature. It seems unlikely that users could inadvertently use or describe the wrong template.

AR: We overall agree with the referee's assessment and appreciate the praise.

R#1.7: Suggested Improvements

Ln 675: TemplateFlow can allow users to calculate 'robust' transformations between any pair of adult human template spaces: (easy) inter-template registration is very useful. The registration uses 10 images from MNI-152 via ANTs. This is described very briefly in the manuscript – more information would be helpful, especially for those who have not used ANTs before.

- AR: The reviewer is right, and to keep the manuscript's scope within the parameters of a resource article, we did not cover this methodological aspect of the work in depth. In order to preempt future readers from getting confused by this feature, we have moved the description to the Supplementary Materials document and increased the level of detail.

R#1.8: Suggested Improvements

Currently, the Archive consists of MRI templates, but could potentially include other modalities. It would be very interesting to see some other modalities and this would increase the scope of the manuscript.

- AR: We thank the reviewer for this suggestion. Amongst many new resources that have been made available within the *TemplateFlow Archive*, we have added PET and SPECT templates currently distributed with SPM12. We have also edited the manuscript accordingly, removing the restriction to MRI from the *Limitations* stated in the Discussion, and including the following text (ll. 184–190):

The only requirement for feature averages and atlases that share a unique template identifier is that they must be spatially in register, regardless of the data sampling strategy (i.e., volume, surface, or mixed). Although the most widely used templates generally comprehend MRI-derived features, *TemplateFlow* is not limited to any specific set of modalities.

R#1.9: Suggested Improvements

TemplateFlow uses lazy loading. After a resource has been requested once, it is then cached. Although users are probably not going to access so many templates that storage is an issue but: can the cache be cleared?

- AR: We thank the reviewer for this useful suggestion. We have updated the *TemplateFlow Client* with the utility of clearing any cached resources at the request of the user. Because the *Client* already had the capability implemented, this only required exposing it to users through the API. The new feature is correspondingly documented and available through `templateflow.conf.wipe()`.

R#1.10: Suggested Improvements

The topic model analysis is interesting but feels like it could be a starting place, not at the end of the Results. Perhaps, it could be moved to Supplemental and mentioned in the Introduction as evidence for the tendency for template choice to be a function of software defaults?

- AR: We agree with the reviewer on this suggestion on the role of the topic model analysis. We also agree that a more hypothesis-oriented analysis would fall beyond the scope of the paper, and therefore

placing it within the Supplemental Materials seems a reasonable option. We have moved the section accordingly, and edited the manuscript with a mention in the Discussion section.

R#1.11: References and clarity

*Yes, previous work and software are well referenced.
The Abstract, Methods, Results and Discussion are generally quite clear. The motivation for the toolbox is described well. Some minor comments.*

AR: We thank the reviewer for the kind words.

R#1.12: Clarity and context

The Introduction could be better structured. There are some long paragraphs that could be split up (e.g. start new paragraph on line 36 'On account. . .')

AR: We thank the reviewer for raising this flag, and have thoroughly re-worked the structure of the manuscript in depth. We have simplified the introduction moving the discussion about methodological variability to a more appropriate point in one of the Results' subsections. This aspect is also briefly revisited in the Discussion.

R#1.13: Clarity and context

Results (ln 137). 'FAIR' is in the title of the manuscript but the FAIR principles themselves are only listed in a supplemental box, the could be moved to the main text. The results could also work though FAIR more systematically (one paragraph per principle?).

AR: We thank the reviewer for these suggestions. We have moved the Supplementary Box 1 listing the FAIR principles to the main text. We have deeply reworked the first subsection of the Results in a way we believe is more aligned with this reviewer's expectations.

R#1.14: Clarity and context

In the Discussion, the authors suggest that one limitation of TemplateFlow is increased potential for methodological flexibility. But this is not a limitation of the toolbox. Presumably, users could now preregister the exact template that they plan to use?

AR: We thank the reviewer for this excellent suggestion. Indeed, some critics of TemplateFlow have indicated the increased methodological flexibility as a limitation of the resource. However, we completely agree with the reviewer on this point. We have made this suggestion part of the revised manuscript, particularly at lines 320–323.

2. Reviewer #2

R#2.1: General Remark

The current work provides general background on the use of anatomical MRI atlases in neuroimaging research, as well as some of the common issues surrounding misreporting and replication difficulties that arise due to differences in "default" atlases provided by popular neuroimaging software. The tool presented in this paper, TemplateFlow, seeks to mitigate these issues by providing a database of common atlases, a python "client" for the integration of TemplateFlow into preprocessing pipelines, thorough atlas file metadata for precise reporting and ease of replication, and a "manager" for users to upload new atlases which are then peer reviewed before being added to the database.

This manuscript is generally well written, clearly outlines the need for a tool such as TemplateFlow, and is accompanied by in-text figures with example code and jupyter notebooks that are available online. The provided examples clearly walk potential users through the basics of using TemplateFlow in a way that is easy to follow and understand, given the user has a very basic knowledge of python. While seasoned fMRI researchers are used to downloading different atlases and hunting down the technical specifics in supplemental materials for use and manuscript reporting, TemplateFlow can greatly mitigate mistakes, misreporting, and also provides clear information for trainees with limited background in this area. I feel that tools like TemplateFlow and manuscripts such as this that outline methodological pitfalls and provide a clear and valid solution are vital for trainees and the development of robust replication practices in our field. I enthusiastically support this work being accepted for publication after the authors address the minor issues outlined below.

AR: We thank the reviewer for the thorough assessment, which is completely aligned with the expectations we outlined for this paper.

R#2.2: (minor)

The jupyter notebook that is hosted online worked flawlessly, was clear, and gave a nice walk-through of what the different TemplateFlow functions were doing and how they could be integrated into a preprocessing pipeline. However, when working through some of the practical examples in the manuscript, there were some typos or errors that I feel should be corrected or updated for clarity or the benefit of less-seasoned readers.

AR: We thank the reviewer for the kind words, and appreciate the suggestions given below.

R#2.3: (minor)

1. I tested the installation and use of the python package locally and the majority of tests were without error. However, I could not follow the example in Figure 5. The get command would return an empty value when following the example that was supposed to pull an atlas while providing filter specifics about the template (desc, resolution, etc). I then slowly tested and removed fields. This function only pulled files and returned a non-empty variable when I provided only the identifier/name, but then it started pulling all MNI152Nlin6Asym riftis (quite a few) and I cancelled the downloads.

I tested this a second time, and this time I was confused whether it was returning an empty variable because I had just previously started downloading all MNI152Nlin6Asym files and the file already existed locally, or if the example code, in fact did not work. If this issue that I encountered while following the text in figure 5 is an error, I would request it is updated so this step works and is easy to follow and replicate the outputs shown in the figure. If this was a case of user error, I would guess that a new trainee would be confused as well, and some clear explanation of why following the example would result in what I experienced should be included.

AR: We thank the reviewer for pointing out this bug. We have fixed this problem, and now extensions with and without leading dot character "." are interpreted equally.

R#2.4: General Remark

Moreover, there are some overstatements and inaccuracies throughout the manuscript which would warrant extensive rewriting in some parts.

AR: We have extensively edited the manuscript in accordance with the suggestions of the reviewers and we hope these inaccuracies and overstatements have been addressed as a result.

R#2.5: (minor)

Supp BoxS2 (minor edit) In the 4th example:

```
>>> print(get_citations("UNCInfant", bibtex=True)[0])
```

This should have:

```
from templateflow.api import get_citations
```

at the top. I was able to assume this and follow along (even with the missing dependency error that I assume is the reason for the warning that the output is for demonstration only). However, for a new trainee this paper could be a great learning source if it was easy to follow along with all examples without errors. (even if outputs aren't correct or fully implemented yet). This is a very minor edit, but one that I would have appreciated when I was an RA and first learning python.

AR: We appreciate the correction and have edited Supp Box S2 accordingly.

3. Reviewer #3

R#3.1: General Remark

This paper describes a resource for archiving, maintaining, and disseminating templates and atlases for the neuroimaging community. To this end, the authors define a series of rules, meta structures, and data structures to systematically and uniformly organize these templates/atlas. Although this resource could provide a new way to organize, document, and disseminate popular templates/atlas, it is unlikely to create broad and significant impact especially in view of existing

and popular resources such as NIH-supported Neuroimaging Tools and Resources Collaboratory (NITRC, <https://www.nitrc.org>). NITRC is equipped with cloud-based computational and storage resources and covers, among many other things, atlases/templates and computational tools.

AR: We thank the reviewer for the time dedicated to reviewing our manuscript. We believe the general remarks given here misrepresent the manuscript and draw an oversimplified picture of *TemplateFlow*. Acknowledging that this could be a consequence of an unclear manuscript, we have deeply reworked the delivery of the contents of the Results section. The titles are now direct statements of what the proposed resource achieves in four aspects: the Archive (covering FAIR and BIDS), the Client (covering access and related issues), maintenance/management (including the Manager), and reproducibility. The bodies of these sections have also been profoundly edited to address this general comment.

Since the only competing resource mentioned by the reviewer is NITRC, we address it specifically with the response to the corresponding specific remark by this reviewer below (see 3.15). As mentioned on that response, we have added a Supplementary Table to the paper making a head-to-head comparison with NITRC.

R#3.2: General Remark

Taken together with the issues listed below, this work, while not without merits, is not sufficient for Nature Methods. A revision is not likely to change this outcome.

AR: Suggesting that the work is scientifically sound (at the least, “not without merits”) but should be submitted elsewhere (where the work can be deemed “sufficient”) is not actionable feedback. As stated in the response to Comment 3.1 above, we have nonetheless made a thorough revision to make the paper more accessible to a broader readership that may be less familiar with the relevance of FAIR availability of strategic research resources such as templates and atlases for neuroimaging.

R#3.3: General remark

The motivation for this work as claimed by the authors is significant. However, this work does not solve the more pressing issue of methodological variability across studies.

AR: The reviewer is right; this manuscript presents a resource and resolving the issue of methodological variability falls outside of the scope of the work as mentioned in Limitations. However, as part of the deep restructuring of the Results section, the last subsection now covers this aspect with increased clarity. Nonetheless, we discuss this issue further in 3.8 below, as that comment more precisely critiques this aspect of the work.

R#3.4: General Remark

*Unlike imaging data themselves, which can be large in terms of scale, there are just a handful of atlases and building an elaborate resource like *TemplateFlow* might be an overkill.*

AR: After careful consideration, we have been unable to understand what this critique means, what evidence it is founded on, as well as the degree to which *TemplateFlow* “might be” an overcomplicated solution to the eyes of the reviewer. Nonetheless, we believe that the revision we are submitting preempts reticent readers from quickly extracting a take like this.

R#3.5: General Remark

The future prospect of TemplateFlow is also not clear, since the number of templates is not expected to grow by much, unlike the imaging data themselves.

AR: The reviewer predicts that the future of *TemplateFlow* is unclear, based on their experience or data we do not have access to. Even if the prediction turned out to be true, the *TemplateFlow Archive* already contains the most widely used templates and atlases used for analysis of neurotypical adult humans. However, we openly disagree with this opinion, and as the manuscript states, *TemplateFlow* is aimed to transform the redistribution model of templates and atlases (currently embedded within software packages, chiefly) into a maximally accessible, self-documented, and, more importantly, community-driven resource.

Since the time these reports were received, *TemplateFlow* has been expanded with several additional, non-human templates (i.e., Mouse In, NMT31Sym, and VALiDATe29), and human templates. Long-term sustainability is ensured, and we have edited the Discussion to reflect this aspect (ll. 396–398).

R#3.6: (major)

Grant information should be included in the metadata of each template resource.

AR: We agree with the reviewer – all templates/atlas should be distributed with such information regardless of the means of distribution (researcher’s website, NITRC, etc.). Templates/atlas missing grant information in their metadata did not have such information in the original site of distribution. Nonetheless, we have contacted several of the authors of templates currently available on the resource and included this information whenever it was possible.

R#3.7: (major)

It is unclear how TemplateFlow manages variations of templates, e.g., ICBM 2009 (a,b,c) Nonlinear (A)Symmetric. For example, will the variations be grouped under a single template resource?

AR: We agree with the reviewer that this is an important consideration. We have added ICBM 2009 (a, b) Nonlinear (A)Symmetric to *TemplateFlow*. Table 1 summarizes that each of the ICBM versions are considered an independent template, and the fact that all these templates represent the same anatomy is encoded by the fact that spatial transforms within the group correspond to the identity transform.

R#3.8: (major)

The improvement of the analytical flexibility across disciplines as the authors claimed is not supported. The current results do not give enough insight into how TemplateFlow can improve analytical flexibility.

AR: We thank the reviewer for offering this possible reading of our paper. Amongst the substantial edits of the Results section, we have dedicated one subsection (entitled “*Decoupling software and templates is indispensable for more reliable and more reproducible study designs*”) to better address this issue.

R#3.9: (major)

It is unclear how TemplateFlow can keep the templates up to date. This issue becomes increasingly critical as the number of templates grows.

AR: We thank the reviewer for letting us know about this unclear point. We have substantially edited the Results subsection that described the management of the resource to address this issue, under the new title “TemplateFlow eases the dissemination of templates and opens their vetting and maintenance to the community”.

R#3.10: (major)

There is no incentive for the template creators to upload their templates to TemplateFlow, more so when it comes to future maintenance. Most people would just download the latest templates through official channels provided by the template developers.

AR: We thank the reviewer for expressing their confusion in this regard. Alongside edits towards addressing 3.9, the new Results subsection now describes more specifically why we would disagree on this statement.

R#3.11: (major)

The authors mentioned integration with fMRIPrep with TemplateFlow. But how will TemplateFlow integrate with other processing pipelines, such as FreeSurfer, FSL, and ANTs, which are more geared towards processing of structural MRI data.

AR: We thank the reviewer for this question. As clearly stated in the manuscript, adoption is intended in the opposite direction: pipeline and tool developers may integrate *TemplateFlow* through its thoroughly described API (application programming interface).

R#3.12: (major)

Surface templates cannot be found on the TemplateFlow website.

AR: We understand the reviewer is referring to the *TemplateFlow Web Browser* at <https://templateflow.org/browse>. The browser is automatically re-generated and posted with each update of the Archive. We have checked and indeed the two surface templates currently available on the Archive are listed and their resources browseable. We have additionally added the NMT31Sym template, which also contains surface data.

R#3.13: (major)

“The quality of peer reviewed template resources is assessed once prior to publication ...”. Why is this considered a poor resource adoption? The templates submitted to TemplateFlow are not going to be reviewed again to suggest changes. It is not clear how TemplateFlow will actually be conducive to the improvement of the templates.

AR: We thank the reviewer for alerting of this possible reading of our manuscript. We now have clarified that we propose *TemplateFlow*'s review process as a complement to the assessment held during

traditional peer-review (ll. 247–249), instead of rendering the point as a criticism to resource adoption. We have also explained why *TemplateFlow* will incentivize and support the improvement of templates in ll. 257–262.

R#3.14: (major)

The relevance of topic modeling analysis is unclear.

AR: We thank the reviewer for raising this issue. Following the more concrete suggestions in this regard by R#1 (see 1.10), we have moved the topic modeling analysis to the Supplementary Materials.

R#3.15: (major)

Unclear what benefits TemplateFlow brings in comparison with popular platforms such as NITRC. Will the benefits bring significant impact to the field not already fulfilled by NITRC?

AR: We thank the reviewer for suggesting a head-to-head comparison with NITRC, which we have added to the paper as Supplementary Table S5. We have also introduced some discussion around issues we have encountered when trying to access templates distributed with NITRC (ll. 388–396). One of those issues makes reference to this repository: <https://www.nitrc.org/projects/neurodevdata>, which is presented as open under the CC-BY-NC, but it is inaccessible. The download button leads to a form that the managers of the resource use to send a DUA along. The DUA disallows sharing the resource, thereby voiding the CC-BY-NC terms. Although we agree that NITRC has been an early and enabling resource for neuroscience, we would argue that it does not cover this particular application with sufficient granularity.

[REDACTED]

Decision Letter, first revision:

Dear Oscar,

Thank you for submitting your revised manuscript "TemplateFlow: FAIR-sharing of multi-scale, multi-species brain models" (NMETH-BC46768B). It has now been seen by the original referees and their comments are below. The reviewers find that the paper has improved in revision, and therefore we'll be happy in principle to publish it in Nature Methods, pending minor revisions to satisfy the referees' final requests and to comply with our editorial and formatting guidelines.

Please note that the acceptance is conditional on the reformatting of the manuscript into a Brief Communication. We can be somewhat flexible on the word count. 1800 words and two relatively small figures would be fine. Also, there are essentially no restrictions on supplementary material.

TRANSPARENT PEER REVIEW

Thank you again for your interest in Nature Methods Please do not hesitate to contact me if you have any questions.

Best regards,
Nina

Nina Vogt, PhD
Senior Editor
Nature Methods

ORCID

Reviewer #1 (Remarks to the Author):

I have re-reviewed the revised manuscript. The authors have made several edits and updates that have addressed my comments. Briefly, these edits include a streamlining of the Introduction, repositioning of tangential material (e.g., the topic model analysis) into the supplemental material and inclusion of FAIR principles in the main text. The updated manuscript is clearer and I thank the authors for their responsive edits.

Reviewer #2 (Remarks to the Author):

I appreciate that the authors thoughtfully and appropriately addressed my concerns (as well as the concerns of the other reviewers). I feel my general notes on the manuscript were addressed thoroughly and the typos or code-specific issues I experienced were corrected. I think that future work in our field will greatly benefit from this manuscript and the tool provided, and I have no further notes or concerns that should delay advancement to publication.

Reviewer #3 (Remarks to the Author):

The revision has raised more concerns:

1. I agree with the observation of Reviewer 1: “In theory, users can obtain, curate and document use of these templates without using TemplateFlow. The novelty of this toolbox is it provides easier access to a range of templates and to better document the specific templates used.”

However, is making already accessible templates easier to access a sufficiently significant contribution to warrant publication in Nature Methods? Templates less accessible due to restrictive licenses, especially when intended by the template authors, remain that way. There is nothing TemplateFlow can do if the template creators want the license to remain restrictive. If a template creator has no intention of complying with the licensing terms of TemplateFlow, there is simply no motivation or incentive to use TemplateFlow as a means of template distribution. To obtain the newest templates, one would visit the template creators’ official distribution sites and not a third-party site like TemplateFlow.

2. “TemplateFlow’s pipeline for submission of new templates integrates peer-review with minimal technical overhead. This review process is proposed as a complement to the traditional assessment of template resources prior to publication, in which reviewers and editors focus on academic merit over accessibility and reusability potential.”

The peer-review process for submitted templates remains unclear and therefore it is uncertain how it

would “complement” traditional assessments. Would a template ever be rejected? Rejected templates would be posted somewhere else by the creators, implying that researchers will still need to search for templates elsewhere as not all templates will be available in TemplateFlow.

3. “To integrate template resources into neuroimaging workflows, traditional approaches required deploying an oftentimes voluminous tree of prepackaged data to the filesystem.”

There are only a limited number of templates available or suitable for a single study, so storage is not going to be an issue. It will not be an issue even if all templates currently at TemplateFlow, which are very limited, need to be downloaded. This is unlike downloading hundreds or thousands of images for a study.

4. “Applied to the particular choice of template and atlas combinations, it would thus be desirable to report neuroimaging results with reference to several standard spaces and determine whether the interpretations hold across those references and atlases.”

In a single study, reporting results for multiple standard spaces is infeasible since it is not often that multiple template creators create different templates for a particular population. Each template is in a sense unique. Using a template not targeted at the population under investigation is simply poor experimental design - a problem TemplateFlow cannot solve.

5. “....., which we have added to the paper as Supplementary Table S5.”. Where is the Supplementary Table S5? Should be Supplementary Table S4?

6. Table S4: “Overall, NITRC is more of a catalog of neuroimaging resources and a community forum than an infrastructure to develop and maintain a technological resource.”

It is unclear how TemplateFlow develops the “technological resource”, i.e., the templates. The template creators develop and maintain the templates; TemplateFlows pull the templates together and redistribute them. In this sense, TemplateFlow is a “catalog” of templates - a function largely fulfilled by NITRC.

7. Unclear why the authors are certain that the mission and design of NITRC “do not accommodate the particular needs of template authors.” (Caption, Table S4). What are “the particular needs of template authors”?

8. “TemplateFlow supports a multifaceted insight into brains across species, and enables multiverse analyses”.

TemplateFlow falls short of realizing multiverse analysis. There are only a handful of atlases in each category. Multiverse analysis only makes sense if multiple templates are available for a specific population of subjects. Multiverse analysis might be an ideal for the future. But this remains a speculation not in any way proven in this work.

9. TemplateFlow currently only contains 25 atlases. Is TemplateFlow an overkill? The number of atlases is not likely to grow quickly as significant efforts and expertise are needed to construct atlases.

10. Supplementary Table S4 “Head-to-head comparison of TemplateFlow and NITRC”:

a) Why is “version control” better than “snapshots”? Templates/Atlases are normally in binary format, not text format. Data diffs cannot be easily and meaningfully visualized.

b) I do not see the need for an API or a client. Downloading templates is relatively uncomplicated and typically needs to be done only once per study since the templates do not change often. Users savvy enough to program with the API would presumably have no problem downloading the templates.

c) Licensing: This, again, is an issue TemplateFlow cannot solve.

11. “Following the FAIR Principles, TemplateFlow effectively decouples standardized spatial data from software libraries while affording processing and analysis workflows (e.g. Esteban et al., 2017, 2019) with the necessary flexibility to select the most appropriate template available.”

I would argue that decoupling templates from their originally intended software packages is not necessarily the right thing to do. Computational tools are often designed based on a template and template-software decoupling would potentially make the problem of methodological variability worse.

Author Rebuttal, first revision:

However, is making already accessible templates easier to access a sufficiently significant contribution to warrant publication in Nature Methods? Templates less accessible due to restrictive licenses, especially when intended by the template authors, remain that way. There is nothing TemplateFlow can do if the template creators want the license to remain restrictive. If a template creator has no intention of complying with the licensing terms of TemplateFlow, there is simply no motivation or incentive to use TemplateFlow as a means of template distribution. To obtain the newest templates, one would visit the template creators' official distribution sites and a not third-party site like TemplateFlow.

AR: We appreciate the time devoted to writing a referee report. Unfortunately, we cannot respond to the initial question of this critique. Indeed, only the editors of Nature Methods can respond to it. The rest of the critique does not offer any actionable suggestion to improve our manuscript nor the tool itself. As clearly stated in the manuscript, *TemplateFlow* does only redistribute resources with permissive licenses. Resources cannot be accepted in the Archive otherwise. Nonetheless, we have reached out to some template authors suggesting license changes or clarifications, which in some cases where the authors accepted, allowed their redistribution with the tool. Regarding the motivation of authors to redistribute their templates via *TemplateFlow* and how users may access them, we refer the reviewer to our previous rebuttal letter.

R#3.2:

2. "TemplateFlow's pipeline for submission of new templates integrates peer-review with minimal technical overhead. This review process is proposed as a complement to the traditional assessment of template resources prior to publication, in which reviewers and editors focus on academic merit over accessibility and reusability potential."

The peer-review process for submitted templates remains unclear and therefore it is uncertain how it would "complement" traditional assessments. Would a template ever be rejected? Rejected templates would be posted somewhere else by the creators, implying that researchers will still need to search for templates elsewhere as not all templates will be available in TemplateFlow.

AR: The reviewer raises concerns regarding how *TemplateFlow's* peer-review process of new template submissions would complement to the traditional assessment of template resources prior publication. The reviewer shapes their concerns with one exemplary question: "Would a template ever be rejected?" As opposed to conventional pre-publication peer-review, the revision by peers of the new resource will ensure that they meet *TemplateFlow's* quality requirements, without evaluating the scientific scholarship of the submission, which we consider addressed by traditional publication. In other words, *TemplateFlow's* review process is more similar to the screening of a manuscript before being posted on a pre-print archive. Further differences with respect to traditional publication include post-publication peer-review, which is built-in in the *TemplateFlow's* workflow. We have edited minimal parts of our manuscript to improve the clarity of this point.

R#3.3:

3. "To integrate template resources into neuroimaging workflows, traditional approaches required deploying an oftentimes voluminous tree of prepackaged data to the filesystem."

There are only a limited number of templates available or suitable for a single study, so storage is not going to be an issue. It will not be an issue even if all templates currently at TemplateFlow, which are very limited, need to be downloaded. This is unlike downloading hundreds or thousands of images for a study.

AR: There is and has never been any claim in the manuscript, as the reviewer implies, that hosting all templates locally would create a storage issue. This comment reflects a misunderstanding of the purpose of a very specific feature of the Python client, in this case lazy loading. Because there is no actionable feedback that we may use to clarify the manuscript or improve the tool, we refer the referee to the comprehensive documentation of *TemplateFlow* we offer online.

R#3.4:

4. *“Applied to the particular choice of template and atlas combinations, it would thus be desirable to report neuroimaging results with reference to several standard spaces and determine whether the interpretations hold across those references and atlases.”*

In a single study, reporting results for multiple standard spaces is infeasible since it is not often that multiple template creators create different templates for a particular population. Each template is in a sense unique. Using a template not targeted at the population under investigation is simply poor experimental design - a problem TemplateFlow cannot solve.

AR: We agree with the reviewer that “Using a template not targeted at the population under investigation is simply poor experimental design” is a problem that *TemplateFlow* does not solve. The manuscript clearly states this, instead proposing tools that will enable a more reliable investigation of template-effects as well as the multiverse of template spaces. Because there is no actionable feedback that we may use to clarify the manuscript or improve the tool, we are unable to update our submission in this regard.

R#3.5:

5. *“....., which we have added to the paper as Supplementary Table S5.”. Where is the Supplementary Table S5? Should be Supplementary Table S4?*

AR: We thank the reviewer for calling this error to our attention; we have, to the best of our ability, rectified all references to figures and tables in the manuscript.

R#3.6:

6. *Table S4: “Overall, NITRC is more of a catalog of neuroimaging resources and a community forum than an infrastructure to develop and maintain a technological resource.”*

It is unclear how TemplateFlow develops the “technological resource”, i.e., the templates. The template creators develop and maintain the templates; TemplateFlows pull the templates together and redistribute them. In this sense, TemplateFlow is a “catalog” of templates - a function largely fulfilled by NITRC.

7. Unclear why the authors are certain that the mission and design of NITRC "do not accommodate the particular needs of template authors." (Caption, Table S4). What are "the particular needs of template authors"?

AR: We thank the reviewer for calling our attention to these concerns. We have edited the table in question (now Supplemental Table S4) and removed this ambiguous language.

R#3.7:

8. "TemplateFlow supports a multifaceted insight into brains across species, and enables multiverse analyses".

TemplateFlow falls short of realizing multiverse analysis. There are only a handful of atlases in each category. Multiverse analysis only makes sense if multiple templates are available for a specific population of subjects. Multiverse analysis might be an ideal for the future. But this remains a speculation not in any way proven in this work.

AR: We do not claim in our manuscript that *TemplateFlow* realizes multiverse analysis—only that it is a useful aid for facilitating multiverse analysis. Because there is no actionable feedback that we may use to clarify the manuscript or improve the tool, we are unable to update our submission in this regard.

R#3.8:

9. *TemplateFlow* currently only contains 25 atlases. Is *TemplateFlow* an overkill? The number of atlases is not likely to grow quickly as significant efforts and expertise are needed to construct atlases.

AR: Because there is no actionable feedback in this speculative argument, we are unable to update our submission on this regard. Nonetheless, two out of three reviewers of this submission seem confident *TemplateFlow* is a timely resource, implicitly responding that it is not an "overkill".

R#3.9:

10. Supplementary Table S4 "Head-to-head comparison of *TemplateFlow* and NITRC":
a) Why is "version control" better than "snapshots"? Templates/Atlases are normally in binary format, not text format. Data diffs cannot be easily and meaningfully visualized.

AR: The purpose of Supplementary Table 4 is not to argue that one system is better than another but to establish that *TemplateFlow* and NITRC occupy distinct niches in the scientific exchange ecosystem. The main purpose of versioning here is unambiguous reference and retrieval (which is outsourced to a dedicated architecture for improved interoperability), rather than visualization of data diffs. Because there is no actionable feedback that we may use to clarify the manuscript or improve the tool, we are unable to update our submission on this regard.

R#3.10:

b) I do not see the need for an API or a client. Downloading templates is relatively uncomplicated and typically needs to be done only once per study since the templates do not change often. Users savvy enough to program with the API would presumably have no problem downloading the templates.

AR: The reviewer makes an *argumentum ad antiquitatem*, which is founded on two false clauses: (i) the point assumes that downloading templates from whatever source the authors have decided is the best option, a claim which our manuscript provides substantial evidence against; (ii) the past justifications for such procedure are now invalid – *TemplateFlow* leverages new technologies (e.g., Datalad) which did not exist before to ensure that templates and atlases are managed and reused with best practices. Because there is no actionable feedback that we may use to clarify the manuscript or improve the tool, we are unable to update our submission on this regard.

R#3.11:

c) Licensing: This, again, is an issue TemplateFlow cannot solve.

AR: See responses to comments 3.1 and 3.9.

R#3.12:

11. “Following the FAIR Principles, TemplateFlow effectively decouples standardized spatial data from software libraries while affording processing and analysis workflows (e.g. Esteban et al., 2017, 2019) with the necessary flexibility to select the most appropriate template available.”

I would argue that decoupling templates from their originally intended software packages is not necessarily the right thing to do. Computational tools are often designed based on a template and template-software decoupling would potentially make the problem of methodological variability worse.

AR: This comment operates under the same *argumentum ad antiquitatem* logic of comment 3.10. The manuscript is clear on this point, which is discussed even at a greater extent. We fully understand that someone involved in the development of software that redistributes templates might be reluctant to update their tooling to leverage *TemplateFlow*, but will gladly offer the necessary support when they finally do so because we are confident it will become the *de facto* standard for neuroimaging templates. Since there is no actionable feedback that we may use to clarify the manuscript or improve the tool, we are unable to update our submission on this regard.

Final Decision Letter:

14th Oct 2022

Dear Oscar,

I am pleased to inform you that your Brief Communication, "TemplateFlow: FAIR-sharing of multi-scale, multi-species brain models", has now been accepted for publication in Nature Methods. Your paper is tentatively scheduled for publication in our December print issue, and will be published online prior to that. The received and accepted dates will be August 23rd, 2021 and October 14th, 2022. This note is intended to let you know what to expect from us over the next month or so, and to let you know where to address any further questions.

Over the next few weeks, your paper will be copyedited to ensure that it conforms to Nature Methods style. Once your paper is typeset, you will receive an email with a link to choose the appropriate publishing options for your paper and our Author Services team will be in touch regarding any additional information that may be required.

Your paper will now be copyedited to ensure that it conforms to Nature Methods style. Once proofs are generated, they will be sent to you electronically and you will be asked to send a corrected version within 24 hours. It is extremely important that you let us know now whether you will be difficult to contact over the next month. If this is the case, we ask that you send us the contact information (email, phone and fax) of someone who will be able to check the proofs and deal with any last-minute problems.

If, when you receive your proof, you cannot meet the deadline, please inform us at rjsproduction@springernature.com immediately.

Once your manuscript is typeset and you have completed the appropriate grant of rights, you will

receive a link to your electronic proof via email with a request to make any corrections within 48 hours. If, when you receive your proof, you cannot meet this deadline, please inform us at rjsproduction@springernature.com immediately.

Once your paper has been scheduled for online publication, the Nature press office will be in touch to confirm the details.

Content is published online weekly on Mondays and Thursdays, and the embargo is set at 16:00 London time (GMT)/11:00 am US Eastern time (EST) on the day of publication. If you need to know the exact publication date or when the news embargo will be lifted, please contact our press office after you have submitted your proof corrections. Now is the time to inform your Public Relations or Press Office about your paper, as they might be interested in promoting its publication. This will allow them time to prepare an accurate and satisfactory press release. Include your manuscript tracking number NMETH-BC46768C and the name of the journal, which they will need when they contact our office.

About one week before your paper is published online, we shall be distributing a press release to news organizations worldwide, which may include details of your work. We are happy for your institution or funding agency to prepare its own press release, but it must mention the embargo date and Nature Methods. Our Press Office will contact you closer to the time of publication, but if you or your Press Office have any inquiries in the meantime, please contact press@nature.com.

Please note that *Nature Methods* is a Transformative Journal (TJ). Authors may publish their research with us through the traditional subscription access route or make their paper immediately open access through payment of an article-processing charge (APC). Authors will not be required to make a final decision about access to their article until it has been accepted. Find out more about Transformative Journals

To assist our authors in disseminating their research to the broader community, our SharedIt initiative provides you with a unique shareable link that will allow anyone (with or without a subscription) to read the published article. Recipients of the link with a subscription will also be able to download and print the PDF. As soon as your article is published, you will receive an automated email with your shareable link.

Please note that you and your coauthors may order reprints and single copies of the issue containing your article through Springer Nature Limited's reprint website, which is located at <http://www.nature.com/reprints/author-reprints.html>. If there are any questions about reprints please send an email to author-reprints@nature.com and someone will assist you.

Best regards,
Nina

Nina Vogt, PhD
Senior Editor
Nature Methods